# The Effect of Intraoperative Ferric Carboxymaltose in Joint Arthroplasty Patients: A Randomized Trial

**DOI:** 10.3390/jcm8101674

**Published:** 2019-10-13

**Authors:** Hee-Sun Park, Tae-Yop Kim, Ha-Jung Kim, Young-Jin Ro, Hwa-Young Jang, Won Uk Koh

**Affiliations:** 1Department of Anesthesiology and Pain Medicine, University of Ulsan College of Medicine, Asan Medical Center, 88, Olympic road 43-gil, Songpa-gu, Seoul 05505, Korea; heespark@amc.seoul.kr (H.-S.P.); alexakim06@gmail.com (H.-J.K.); yjro@amc.seoul.kr (Y.-J.R.); janghy315@gmail.com (H.-Y.J.); 2Departments of Anesthesiology and Pain Medicine, Konkuk University Medical Center, 120-1 Neungdong-ro, Gwangjin-gu, Seoul 05030, Korea; taeyop@gmail.com

**Keywords:** iron supplementation, ferric carboxymaltose, total knee arthroplasty, total hip arthroplasty, anemia

## Abstract

This study assessed the efficacy of intraoperative high-dose intravenous iron therapy in facilitating recovery from postoperative anemia and reducing the transfusion rate in patients with total knee and total hip arthroplasty. This prospective randomized controlled study involved 58 subjects. Group F received 1000 mg intravenous ferric carboxymaltose and Group C received normal saline. The changes in hemoglobin (Hb), hematocrit, iron metabolism variables, transfusion rates, and the arterial partial pressure of oxygen and the fraction of oxygen (PaO2/FiO2) ratio were recorded. There were 29 patients of each group. The change in Hb levels from baseline to 1 month post-surgery was higher in Group F than in Group C (0.3 ± 1.0 g/dl vs. −0.8 ± 0.8 g/dl, *p* < 0.001). Functional iron deficiency occurred more frequently in Group C (0% vs. 48.3%, *p* < 0.001) after the operation. The incidence of postoperative anemia, transfusion rate and P/F ratio did not significantly differ between the two groups. This study suggests that intraoperative high-dose ferric carboxymaltose during lower limb total arthroplasty can facilitate the recovery from postoperative anemia. Although it could not prevent the occurrence of postoperative anemia or the administration of transfusion, this treatment seemed to overcome surgery-related decrease of iron availability.

## 1. Introduction

Total hip arthroplasty (THA) and total knee arthroplasty (TKA) are associated with considerable blood loss in the perioperative period [1]. This may cause postoperative anemia, which often results in allogenic blood transfusion [2]. Perioperative transfusion is a risk factor for occurrence of many transfusion-related complications [3]. Furthermore, considering the high proportion of elderly patients undergoing lower limb arthroplasty, transfusion-related adverse effects, such as acute lung injury and sepsis [4], can have a greater negative impact in this population [5]. Allogeneic blood transfusion has been associated with a poor postoperative outcome, including hospital stay, infection, morbidity, and mortality. For these reasons, the importance of patient blood managements (PBM) have increased. Iron replacement strategies [6,7,8,9] with or without erythropoiesis-stimulating agents (ESAs) [10] from several weeks before surgery have been found to be effective in reducing the requirement of transfusion through optimizing the hemoglobin (Hb) level and correcting the iron deficiency in patients undergoing orthopedic surgery. However, iron treatment 2–4 weeks before the scheduled day of operation is not always possible. Elective schedules can always be altered for many reasons and it is not possible in case of lower limb fractures requiring an urgent operation.

The application of several types of intravenous iron infusion in the perioperative period has been studied in patients undergoing THA or TKA [11,12,13,14]. Ferric carboxymaltose (FCM) (Ferinject^®^, Vifor Int., St.Gallen, Switzerland) can be given intravenously in high doses up to 1000 mg of iron rapidly over 15 min. Considering other iron preparations, such as iron sucrose, which can be given in a dose of up to 300 mg per day for 1 h [15], FCM is convenient for both physicians and patients without the need for several intravenous injections.

We hypothesized that a single intravenous FCM administration in the intraoperative period could improve the correction of anemia after surgery by facilitating recovery from postoperative anemia. In this prospective randomized controlled study, we studied the effect of an intraoperative single injection of FCM on hemoglobin change, transfusion rate, and improvement of pulmonary oxygenation in patients undergoing primary unilateral TKA or THA. 

## 2. Methods

### 2.1. Study Population and Randomization

Between June 2017 and October 2018, patients scheduled for elective primary unilateral TKA or THA were screened and enrolled. All patient follow-up ended on December 2018. This study was a single-blinded (study patient), Phase IV, randomized, controlled, single-center trial. This study protocol was approved by the institutional review board (IRB) of the study center (IRB number 2016-1198) and registered in Clinical Trials.gov (NCT02544464). This study adheres to the applicable CONSORT guideline.

All patients provided informed written consent to participate in the study. Patients who met all inclusion and none of the exclusion criteria, were enrolled into the study. Eligible patients were randomly assigned into one of two groups through stratified block randomization of a 1:1 ratio. Stratification factors were operation types, because the result of this study may defer among the either TKA or THA. The patients who were randomized to receive FCM were regarded as Group F (FCM group), while patients who were randomized to receive normal saline (control group) were regarded as Group C. The allocation details were concealed in an opaque sealed envelope. After obtaining informed consent from each patient, the envelope was opened by an independent anesthesiologist not involved in this study assessments. Subjects were assigned to either Group F or Group C. Intravenous FCM 1000 mg was infused to Group F after the induction of general anesthesia or performing spinal block prior to skin incision. 1000 mg FCM was diluted in 100 ml 0.9% sodium chloride solution and administered for 15 min. The patients in the control group received 100 ml 0.9% sodium chloride. The patients in both groups did not take oral iron supplementation after operation. 

Inclusion criteria for enrollment were patients undergoing primary elective unilateral TKA or THA who were American Society of Anesthesiologist (ASA) grade 1–3 aged ≥20 or ≤80 years, and able to give informed consent. Eligible laboratory criteria were an Hb level of more than 10 g/dL within 4 weeks before the operation day and patients with serum-ferritin <300 mg/dL (male) or 200 mg/dL (female).

The exclusion criteria were preoperative Hb ≤ 10 g/dL within 4 weeks before operation; chronic pulmonary disease; chronic renal, cardiovascular, or liver disease; malignancy; patients who had a history of known anaphylaxis and sensitivity to any oral or intravenous iron; iron supplement within six months prior to enrollment; and blood transfusion within two months prior to enrollment. Patients who had surgery in distant operating rooms where the authors did not take charge of anesthesia and follow-up were further excluded.

### 2.2. Data Collected and Outcomes Measures

The primary outcome of this trial was to compare the efficacy of intravenous FCM with the control group in TKA or THA patients. The primary efficacy endpoint was the change in Hb and hematocrit (Hct) and its difference between the preoperative baseline to 1 month after surgery (POD30). Secondary objectives were the changes in Hb, Hct, and the difference at the first and fifth day (POD1 and POD5) after operation compared with preoperative baseline values, change of iron metabolism variables including serum iron, ferritin, transferrin iron binding capacity (TIBC), and transferrin saturation (TSAT) before study drug infusion and at POD5. The effect of FCM on perioperative PaO_2_/FiO_2_ ratio improvement compared with Group C was measured. To measure P/F ratio, arterial blood gas analysis (ABGA) was done at the following time points: before the surgery under room air, intraoperative, post anesthetic care unit (PACU), and POD5. The prevalence of related adverse events including hypersensitivity reactions for FCM were also collected.

### 2.3. Anesthetic Management 

The type of anesthesia was selected at the discretion of the attending anesthesiologist. In every case, anesthesia was performed in the standardized manner of our institution. Spinal anesthesia was performed in the lumbar 3–4 or 4–5 interspace in the lateral position. Patients received intrathecal injection of 12–15 mg of hyperbaric bupivacaine with 10–15 µg of intrathecal fentanyl. Oxygen was supplied to patients via facial mask in a flow of 6 L/min. For intraoperative sedation of the patient, 1–3 mg of midazolam and continuous infusion of intravenous dexmedetomidine (0.5–1.0 µg/kg/h with/without the loading dose of 0.5–1.0 µg/kg for 15 min) or target control infusion (TCI) of propofol at effective site target concentration of 1.0–1.5 µg/kg/min using a programmed target concentration infusion device (Orchestra^®^, Fresenius Kabi AG, Bad Homburg, Germany). In case of general anesthesia, anesthesia was induced with 40 mg lidocaine, 1.5–2.0 mg/kg of propofol and 1 µg/kg of fentanyl. Intravenous 0.6–1.0 mg/kg of rocuronium was given to facilitate endotracheal intubation and anesthesia was maintained with 5%–8 vol% of desflurane and 50% of oxygen with nitrous oxide. Mechanical ventilation was applied without positive end-expiratory pressure, using a constant tidal volume of 6–8 mL/kg and a constant end-tidal carbon dioxide tension of 30–35 mmHg. Arterial cannulation was placed in every study patient before the induction of anesthesia. After the surgery, the study subjects were moved to PACU and oxygen was supplied at 6 L/min via facial mask. The vital signs were monitored, and the patients were evaluated with the modified Aldrete scoring system and until a given score of 9 was reached to be transferred from the PACU to the general ward.

The attending anesthesiologist and the surgeon were encouraged to follow the restrictive transfusion triggers of Hb < 8 g/dL throughout the entire perioperative period. If significant hemodynamic instability was observed despite adequate fluid administration or a requirement of an increasing amount of vasopressor was essential, allogenic transfusion of packed red blood cells (RBC) was permitted with the Hb ≥ 8 g/dL. In case of transfusion, the subjects received one unit of packed RBC and the Hb level was followed to assess the further administration of additional packed RBC.

### 2.4. Statistical Analysis and Sample Size

The sample size of this study was determined on the basis of our pilot study (not published), with calculations based on a minimum of a 20% difference in standard deviation and 80% power to obtain a difference of Hb 1.5 g/dL with an estimated standard deviation of 2.5. It was determined that 26 patients were required in each group. We decided to recruit a total of 58 patients with the designated number of 29 patients in each study group.

Data are expressed as mean (± standard deviation) or median [interquartile range (IQR)] as appropriate for continuous variables, while numbers and percentages were used for categorical variables. Continuous variables were compared using Student’s t-test or Mann–Whitney U test after a normality test (Kolmogorov–Smirnov test). Categorical data were compared using χ^2^ and Fisher’s exact tests as appropriate. Repeated measurements of P/F ratio were compared using a repeated measures analysis of variance (ANOVA) to evaluate the interaction of time and treatment between two groups. A *p* value of < 0.05 was considered to be significant. Data were analyzed using the SPSS^®^ version 25 for Windows^®^ program (SPSS, Chicago, Illinois, USA).

## 3. Results

### 3.1. Baseline and Clinical Characteristics

A total of sixty-nine subjects were enrolled and 58 (84.1%) subjects completed the trial in the period between June 2017 and December 2018. The details of patient disposition are summarized in Figure 1. Three patients were withdrawn after subject enrollment, due to uncontrolled hypertension, unexpected cancellation of surgery on the day of operation due to personal reason, and preoperative transfusion. Eight patients were dropped out due to follow-up loss before 1 month after operation (*n* = 5), unnoticed cancer diagnosis (*n* = 1), and patient withdrawal of the consent (*n* = 2). The clinical characteristics of patients and surgical details are shown in Table 1. There was no difference between the two groups.

### 3.2. Primary and Secondary Outcome

The changes of Hb and Hct (ΔHb30 and ΔHct30) between the preoperative baseline and 1 month post-surgery demonstrated statistical significance favoring the group F compared with group C (Figure 2a,b and Table 2, *p* < 0.001). The changes of Hb and Hct between the preoperative baseline values compared with postoperative day 1 and 5 (ΔHb1 and ΔHct5) was not different between the study groups (Table 2). The number of patients presenting postoperative anemia and the perioperative transfusion rate was not significantly different between both study groups (Table 2). The mean amounts of RBC transfusion were 1.5 units of Group F and 2.5 units of Group C.

Although the preoperative iron storage test was not different between the two groups, the level of serum ferritin and TSAT increased significantly in Group F compared with that of group C at POD5, which lead to a significant difference (*p* < 0.001) in the number of patients demonstrating functional iron deficiency anemia between the two study groups (Table 3).

Repeated measures ANOVA revealed no significant interaction between the two groups over time (*p* = 0.878). There was no report of adverse events related FCM infusion during the entire study period.

## 4. Discussion

In this prospective, randomized controlled trial, we evaluated that intraoperative intravenous iron treatment could influence postoperative anemia recovery in patients undergoing TKA or THA. We found that 1000 mg of FCM treatment could accelerate the recovery from postoperative anemia. This finding is supported by a prospective observational study in elective TKA patients that perioperative iron sucrose administration may accelerate the recovery of postoperative anemia [16].

The systematic approach to optimizing Hb and limiting blood loss strategies resulted in lower transfusion rate [5,6,8,9]. Despite iron supplementation having been effective in decreasing the requirement for transfusion in major orthopedic surgery, intravenous iron has been of limited use because of undesirable adverse events and the need to repeated administration. Iron sucrose, a parenteral iron which was used in many studies, can be administered at a dose of 300 mg for one hour with the maximum weekly dose of 600 mg. Therefore, it requires at least two or three visits and injections for the patient before surgery. The study of perioperative (within 1 week of the surgery) and immediate postoperative intravenous iron replacement with or without ESA has also been reported, but most cases used iron sucrose [12,14,16].

Ferric carboxymaltose, the current study drug, is a dextran-free complex that has better safety profiles than other high molecular weight iron dextran. In nonclinical studies, FCM has less oxidative stress on tissue than other iron preparations [17]. These factors lead to FCM being available as a single rapid infusion formula of a large daily dose (up to 1000 mg) in 15 min. As a result, FCM has the advantages of less patient injections and less visits to hospital. High-dose iron treatment has shown benefits compared with iron sucrose in a previous study, where high-dose of iron treatment (FCM) made similar post-treatment Hb levels, but was much more rapid compared to iron sucrose administration in patients with iron deficiency anemia [18]. A previous report that FCM injection within 2–4 weeks before TKA or THA for optimizing preoperative Hb resulted in a reduce transfusion rate and length of hospital stay [19]. There is also a previous observational study reporting that perioperative use of iron sucrose and FCM is associated with reduced transfusion rates and length of hospital stay without increasing morbidity or mortality [11]. However, there are few prospective studies looking at the effect of single intraoperative FCM in lower limb joint arthroplasty patients. Although it cannot substitute the known optimal method for treating iron deficiency anemia preoperatively [20], evaluating the effect of single dose for intraoperative administration of intravenous iron may have clinical importance, as iron supplementation before surgery is not always possible.

Not much is known about the time required for full recovery from postoperative anemia. Previous studies described that patients undergoing orthopedic surgery without iron supplementation regained 78%–84% of their baseline hemoglobin level at postoperative week 3 [21], or two-thirds correction of anemia at six weeks after surgery [22]. In the current study, we compared the changes of Hb and Hct levels between preoperative and postoperative one month values in the two study groups. Indeed, the Hb and Hct levels of patients that received FCM demonstrated more rapid recovery to baseline values in this study, and the control group showed a pattern likely to be similar to the previous studies without iron supplementation’s results shown above [21,22].

The iron store status of patients can influence the recovery of postoperative anemia. The patients of both groups had similar preoperative iron storage without any intestinal disease. Considering this, we could assume that the rapid recovery of Hb and Hct values of the patients in Group F compared to Group C at 1 month after surgery in this study would have been the benefit from the intraoperative parenteral iron administration. A recent prospective randomized controlled study gave a single dose of FCM (800–1000 mg) in patients with functional iron deficiency anemia at POD1, and the mean Hb level demonstrated improvement at 4 weeks after the operation in the FCM group [23]. In this study, Group C has shown a high incidence of functional iron deficiency compared to Group F (48.3% vs. 0%). It demonstrated that intraoperative single dose FCM may prevent or reduce iron deficiency even in patients without preoperative iron deficiency. If we included subjects with preoperative iron deficiency anemia, it would have yielded different results.

According to international consensus statement, if surgical patients have functional iron deficiency or iron sequestration, intravenous iron treatment with or without ESAs is recommended [24]. Postoperative anemia has related functional iron deficiency and iron sequestration which is a situation where although the individual has sufficient iron storage, the stored iron cannot be appropriately utilized for erythropoiesis [25]. Surgery-related inflammation increases hepcidin synthesis, which may reduce iron mobilization from body stores and as a result, inhibit erythropoiesis [22,26]. Postoperative oral iron treatment, however, demonstrated to be not effective in correcting anemia in randomized controlled trials [21,27]. Intravenous iron treatment provides a sufficient refill of iron stores [28] and increases circulating iron, which causes enhanced iron availability [12], and thus hastened recovery of Hb in surgical patients [26,29]. For functional iron deficiency patients, parenteral iron with additive ESAs is an effective useful treatment of anemia.

The present study has several limitations. First, we did not exam the iron metabolism variables tested at 1 month post-surgery. These will give more information for assessing the iron availability and managing postoperative anemia at 1 month postsurgery. If most of the subjects had iron deficiency anemia, this study would yield different results, and 1 month postsurgery iron metabolism variables also would have a clinical outcome. Second, we had small number of patients to compare two groups which may have underpowered the secondary outcome results. A statistical difference in the baseline preoperative Hb and Hct values between the two study groups may have also influenced the negative results of absolute Hb, Hct values, and transfusion rates. Third, we could not evaluate the relationship between postoperative anemia and functional outcome or quality of life after joint arthroplasty. A study with larger population will be needed to further evaluate how short-term administration of ferric carboxymaltose might help in early recovery, length of hospital stay, clinically related outcomes including quality of life, serious adverse events, and safety parameters. Fourth, the requirement of transfusion and postoperative drop of Hb or Hct may have been different between the surgical types, but we performed a stratified randomization depending on the type of surgery. Thus, a similar rate of THA and TKA cases were enrolled at each group which minimized the difference in operation characteristics. Fifth, this study routinely gave 1000 mg FCM which in some patients can be an over-administration of the study drug. A patient’s total body iron deficit (TID) can be calculated using the Ganzoni formula TID (mg) = weight (kg) x (ideal Hb – actual Hb) (g/dL) x 2.4 + depot iron (500 mg) [30]. According to this formula, our study subjects weighing 65 kg with a Hb level 10 g/dL would have a body iron deficit of about 812 mg. This indicates that 1000 mg of FCM in this study was not much of an overdose in our study patients.

## 5. Conclusions

In conclusion, a single injection of FCM 1000 mg intraoperatively can facilitate Hb recovery in patients undergoing unilateral lower limb total arthroplasty and it can be used without adverse events. Intraoperative intravenous iron use did not reduce the incidence of postoperative anemia and transfusion rate, but it help to overcome surgery-related functional iron deficiency and helps iron availability.

## Figures and Tables

**Figure 1 jcm-08-01674-f001:**
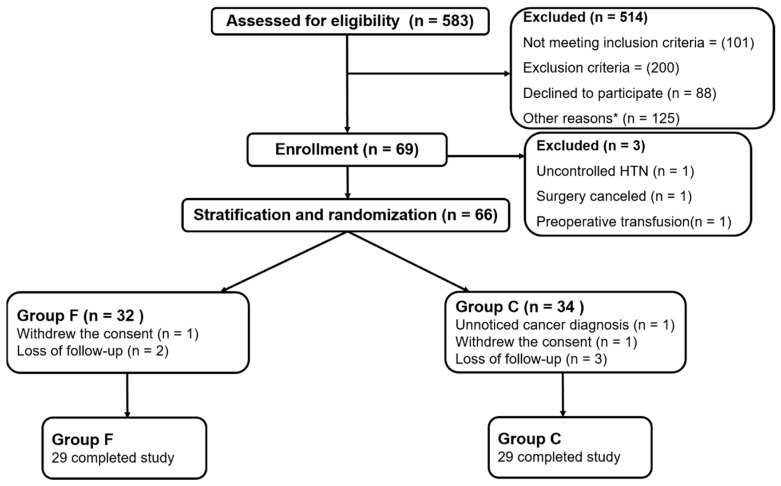
Flow chart of patient enrollment. *Patients who had surgery in operating rooms where the authors did not take charge of anesthesia and follow-up.

**Figure 2 jcm-08-01674-f002:**
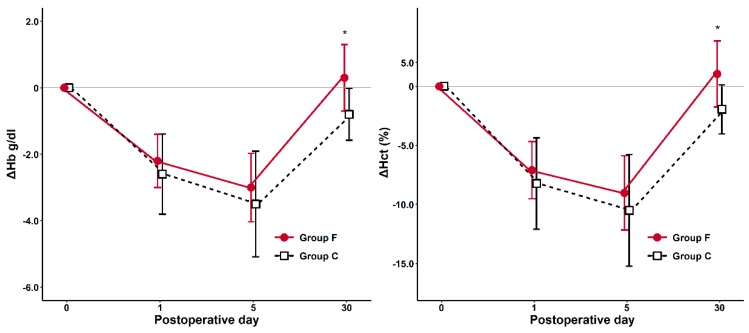
The changes of hemoglobin (ΔHb) (**left**) and hematocrit (ΔHct) (**right**) between baseline and postoperative day in two groups. **p* < 0.05. Hb, hemoglobin; Hct, hematocrit.

**Table 1 jcm-08-01674-t001:** Characteristics and clinical data of study patients in each group.

Characteristic	Group F (*n* = 29)	Group C (*n* = 29)
Age (yr)	65.1 (±10.2)	62.0 (±9.5)
Female	19 (65.5%)	21 (72.4%)
Height (cm)	156.9 (±7.1)	158.2 (±6.4)
Weight (kg)	64.2 (±9.8)	67.1 (±11.8)
BMI (kg/m^2^)	26.2 (±4.0)	26.9 (±4.8)
Diabetes mellitus	4 (13.8%)	5 (17.2%)
Hypertension	13 (44.8%)	14 (48.3%)
Total hip arthroplasty	19 (65.5%)	20 (69.0%)
Total knee arthroplasty	10 (34.5%)	9 (31.0%)
Regional anesthesia	21 (72.4%)	17 (58.6%)
Anesthesia time (min)	164.6 (±23.7)	160.3 (±24.8)
Operation time (min)	136.1 (±21.9)	129.4 (±21.2)
Crystalloid (ml)	950.0 [650.0]	750.0 [600.0]
Tranexamic acid	19 (65.5%)	20 (69.0%)

Data are presented as mean (±standard deviation), median [interquartile range] or *n* (%). Group C, control group; Group F, ferric carboxymaltose group; BMI, body mass index.

**Table 2 jcm-08-01674-t002:** Transfusion rate, the incidence of anemia, hemoglobin, and hematocrit values of each group.

	Group F (*n* = 29)	Group C (*n* = 29)	*p*-Value
Patients transfused	2 (6.9%)	4 (12.8%)	0.670
Postoperative anemia			
POD1	25 (86.2%)	24 (82.7%)	0.717
POD5	28 (96.6%)	26 (89.7%)	0.300
POD30	9 (31.0%)	9 (31.0%)	1.000
Hb (g/dl)			
Baseline	12.5 (±1.3)	13.4 (±1.1)	0.007
POD1	10.3 (±1.4)	10.7 (±1.3)	0.195
POD5	9.5 (±1.2)	9.9 (±1.6)	0.318
POD30	12.8 (±1.3)	12.6 (±1.1)	0.433
ΔHb1	−2.2 (±0.80)	−2.6 (±1.21)	0.133
ΔHb5	−3.0 (±1.03)	−3.5 (±1.59)	0.156
ΔHb30	0.3 (±1.00)	−0.8 (±0.78)	<0.001
Hct (%)			
Baseline	38.1 (±4.0)	40.5 (±3.0)	0.014
POD1	31.0 (±4.2)	32.2 (±4.1)	0.272
POD5	29.1 (±3.8)	29.9 (±4.8)	0.484
POD30	39.2 (±3.9)	38.5 (±3.1)	0.470
ΔHct1	−7.1 (±2.42)	−8.23 (±3.87)	0.190
ΔHct5	−9.02 (±3.14)	−10.5 (±4.73)	0.153
ΔHct30	1.04 (±2.80)	−2.0 (±2.10)	<0.001

Data are presented as mean (±standard deviation) or *n* (%). Group C, control group; Group F, ferric carboxymaltose group; POD, postoperative day; ΔHb1 and ΔHct1, Hb and Hct change between POD1 and baseline; ΔHb5 and ΔHct5, Hb and Hct change between POD5 and baseline; ΔHb30 and ΔHct30, Hb and Hct change between POD30 and baseline.

**Table 3 jcm-08-01674-t003:** Iron metabolism variables.

	Preoperative			Postoperative Day5		
	Group F	Group C	*p*-Value	Group F	Group C	*p*-Value
Ferritin (ng/mL)	90.6 [82.5]	87.0 [105.2]	0.846	1248.1 [494.8]	170.8 [157.2]	< 0.001
TIBC (ug/dL)	287.0 [55.0]	288.0 [49.0]	0.803	220.0 [51.0]	236.0 [58.5]	0.103
Iron (ug/dL)	92.4 [37.0]	101.0 [46.5]	0.146	76.0 [33.0]	30.0 [30.5]	< 0.001
TSAT (%)	29.4 [16.3]	33.2 [17.2]	0.118	38.1 [26.7]	13.7 [12.0]	< 0.001
Absolute iron deficiency	2 (6.9%)	2 (6.9%)	1	0	3 (10.3%)	0.237
Functional iron deficiency	0	0	1	0	14 (48.3%)	< 0.001
CRP	0.10 [0.16]	0.11 [0.15]	0.560	7.49 [2.92]	7.95 [4.63]	0.936

Data are presented as median [interquartile range] and n (%). Group C, control group; Group F, ferric carboxymaltose group; TIBC, total iron-binding capacity; TSAT, transferrin saturation; CRP, C-reactive protein. Absolute iron deficiency anemia: low Hb (men <13 g/dL, and women <12 g/dL) with TSAT <20%, ferritin <30 ng/ml, and no signs of inflammation; Functional iron deficiency: low Hb (men <13 g/dL, and women <12 g/dL) with TSAT <20%, ferritin >100 ng/mL, and CRP >5 mg/dL.

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
