# Peer review of "The Effect of Intraoperative Ferric Carboxymaltose in Joint Arthroplasty Patients: A Randomized Trial"

_jcm, 2019, doi:10.3390/jcm8101674_

Round 1
Reviewer 1 Report
Recently, new formulations for parenteral iron have reduced time and resource requirements for its administration, enabling pragmatic treatment strategies for patients with iron deficiency undergoing surgery. The current study assesses a single short-term infusion of ferric carboxymaltose following induction of anesthesia, versus a saline placebo, for the correction of anemia after surgery in patients undergoing primary unilateral total knee or hip arthoplasty. This reviewer thanks the authors for undertaking this important randomized controlled trial addressing feasible treatment strategies for perioperative anemia and transfusion avoidance, and recommends the following suggestions improve the strength of this manuscript:
Methods and Results:
This reviewer noticed significant methodological discrepancies between this manuscript and its corresponding clinicaltrials.gov registration.
1) Could the authors please update their clinicaltrials.gov registration to reflect the current trial, particularly for the primary and secondary outcomes.
2) The clinicaltrials.gov application indicates that the trial was to be conducted in a triple-blinded fashion (the participant, investigator, and outcomes assessor masked from allocation); however the manuscript indicates that this was a single-blinded study (participant masked from allocation). Could the authors please elaborate why the study investigators/staff were not blinded to treatment allocation?
In Figure 1, Of the 514 participants excluded prior to enrollment, a large proportion (~25%; n=125) were excluded for ‘other reasons’. Could the authors please elaborate on the other reasons for exclusion for these patients.
It is unclear what is meant by the last statement of section 2.1 Study population and randomization: “Furthermore, patients with other conditions which according to the opinion of the investigator were considered unacceptable were excluded from enrollment.”
Could the authors please provide a definition or description of ‘unacceptable conditions’ in this context.
It is unclear what is meant by the last statement of section 2.3 Anesthetic management: “The patient response to transfusion as considered positive when the level of Hb increase was ≥ 1.5g/dl.” Therefore, are the transfusion values in Table 2 in reference to all patients transfused, or to those who had a positive transfusion response?
- Furthermore, it would be beneficial to report the number of units transfused per transfused patient when characterizing the efficacy of short-term ferric carboxymaltose.
Discussion:
This reviewer recommends that it should be emphasized that this approach should not be substituted for optimal methods for treating iron deficiency anemia pre-operatively (eg Kei et al CJA 2019).
In the limitations section, the authors should suggest a larger study should evaluate clinically related outcomes; quality of life, serious adverse events; and safety parameters associated with short-term administration of ferric carboxymaltose in this patient population.
The phrase ‘need for transfusion’ should be replaced with ‘receipt of transfusion’ or ‘administration of transfusion’ since there is still considerable debate as to when a transfusion is ‘needed’
Please clarify exactly when the study drug was administered. The manuscript indicates it was given “after the induction of anesthesia prior to skin incision” which is terminology usually associated with general anesthesia. Since all patients had spinal anesthesia, might this better be worded as something like “after the induction of general anesthesia or performance of spinal block but prior to skin incision”
For the sample size calculations, did the phrase “a difference of 1.5 g/dl” refer to hemoglobin concentration? If so, please revise the sentence.
The term ‘perioperative’ is frequently used for either preoperative, intraoperative and/or postoperative period. It would seem that the iron in this study was actually given intraoperatively. If that is the case, perhaps the wording should be changed accordingly.
Author Response
Response: JCM-587876, entitled “Effect of intraoperative ferric carboxymaltose in joint arthroplasty patients: a randomized trial”
Dear Editor
We would like to thank the editor and reviewers for granting us the possibility to resubmit our manuscript and for their thoughtful reviews. We appreciate truly their time and efforts on our manuscript. Based on reviewer’s advice and suggestion, we have revised the manuscript to the best of our abilities. Below, we respond to their remarks (in bold) and include direct quotes from the manuscript (in red) point-by-point.
Best regards,
*Corresponding Author
Won Uk Koh
Department of Anesthesiology and Pain Medicine, Asan Medical Center, University of Ulsan College of Medicine, 88 Olympic-ro 43-gil, Songpa-gu, Seoul 05505, Republic of Korea
Tel: +82-2-3010-5606 Fax: +82-2-3010-6790
Email: koh9726@naver.com
Reviewer #1.
Recently, new formulations for parenteral iron have reduced time and resource requirements for its administration, enabling pragmatic treatment strategies for patients with iron deficiency undergoing surgery. The current study assesses a single short-term infusion of ferric carboxymaltose following induction of anesthesia, versus a saline placebo, for the correction of anemia after surgery in patients undergoing primary unilateral total knee or hip arthoplasty. This reviewer thanks the authors for undertaking this important randomized controlled trial addressing feasible treatment strategies for perioperative anemia and transfusion avoidance, and recommends the following suggestions improve the strength of this manuscript:
Methods and Results:
This reviewer noticed significant methodological discrepancies between this manuscript and its corresponding clinicaltrials.gov registration.
1) Could the authors please update their clinicaltrials.gov registration to reflect the current trial, particularly for the primary and secondary outcomes.
è Thank you for your kind comments. We updated the contents of ‘Clinical Trials.gov’ to reflect the current trial. We are waiting for ‘Clinical Trials.gov’ to approve these modifications. Furthermore, we added in the abstract, method and results section about the improvement of pulmonary oxygenation after IV iron treatment. It did not show statistically significant results. (line 19-20, line 59, line 102-105, line 146-148, line 178
2) The clinicaltrials.gov application indicates that the trial was to be conducted in a triple-blinded fashion (the participant, investigator, and outcomes assessor masked from allocation); however the manuscript indicates that this was a single-blinded study (participant masked from allocation). Could the authors please elaborate why the study investigators/staff were not blinded to treatment allocation?
è In this study, we used a single-blinded method because of ferric carboxymaltose can be easily identified by the medical staff and the investigator due to its red color. Per your recommendation, we revised masking method from triple to single.
In Figure 1, Of the 514 participants excluded prior to enrollment, a large proportion (~25%; n=125) were excluded for ‘other reasons’. Could the authors please elaborate on the other reasons for exclusion for these patients.
It is unclear what is meant by the last statement of section 2.1 Study population and randomization: “Furthermore, patients with other conditions which according to the opinion of the investigator were considered unacceptable were excluded from enrollment.” Could the authors please provide a definition or description of ‘unacceptable conditions’ in this context.
è We apologize this sentence can confuse reviewers. The main ‘other reasons’ were that these surgeries took place in another operating room where our authors did not charge of anesthesia and patient follow-up. In method section, we revised and now it reads “Patients who had surgery in distant operating rooms where the authors did not take charge of anesthesia and follow-up were further excluded”in line 91. We have also added footnote of Figure 1 as ‘*Patients who had surgery in operating rooms where the authors did not take charge of anesthesia and follow-up’.
It is unclear what is meant by the last statement of section 2.3 Anesthetic management: “The patient response to transfusion as considered positive when the level of Hb increase was ≥ 1.5g/dl.” Therefore, are the transfusion values in Table 2 in reference to all patients transfused, or to those who had a positive transfusion response?
è We deleted this sentence, as it may give inappropriate confusion to the readers. Table 2 shows just the number of patients who received red blood cell (RBC) during intra- and post-operative period, not a positive transfusion responses. Further, we miscounted the number of transfused patients and revised it in Table 2.
It would be beneficial to report the number of units transfused per transfused patient when characterizing the efficacy of short-term ferric carboxymaltose.
è Thank you for your suggestion. Our subjects received RBC from 1 unit to 3 units. We referred the amount of RBC unit in Result section, as “The mean amounts of RBC transfusion were 1.5 units of Group F and 2.5 units of Group C.”
Discussion section :
This reviewer recommends that it should be emphasized that this approach should not be substituted for optimal methods for treating iron deficiency anemia pre-operatively (eg Kei et al CJA 2019).
We agree your comments. Intraoperative intravenous iron treatment cannot be a substitute for optimal preoperative methods for iron replacement for treating iron deficiency anemia. We have added the following sentence in the discussion section line 231-232. “Although it cannot substitute the known optimal method for treating iron deficiency anemia preoperatively, ”
In the limitations section, the authors should suggest a larger study should evaluate clinically related outcomes; quality of life, serious adverse events; and safety parameters associated with short-term administration of ferric carboxymaltose in this patient population.???
è Thank you for the comment. We agree that a study with a larger population is needed to further evaluate the clinically related outcomes. We have added the following sentence in the discussion section 273-275. A study with larger population will be needed to further evaluate how short-term administration of ferric carboxymaltose might help in early recovery, length of hospital stay, clinically related outcomes including quality of life, serious adverse events and safety parameters.
The phrase ‘need for transfusion’ should be replaced with ‘receipt of transfusion’ or ‘administration of transfusion’ since there is still considerable debate as to when a transfusion is ‘needed’
è Thank you for your suggestion. We changed the phrase ‘need for transfusion’ to ‘the administration or the requirement of transfusion; in Abstract, Introduction, Method 2.3, and Discussion section. Thank you for your kindly comments.
Please clarify exactly when the study drug was administered. The manuscript indicates it was given “after the induction of anesthesia prior to skin incision” which is terminology usually associated with general anesthesia. Since all patients had spinal anesthesia, might this better be worded as something like “after the induction of general anesthesia or performance of spinal block but prior to skin incision”
è We corrected when the study drug was injected in Method sections, as “Intravenous FCM 1000 mg was infused to Group F after the induction of general anesthesia or performing spinal block prior to skin incision.”
For the sample size calculations, did the phrase “a difference of 1.5 g/dl” refer to hemoglobin concentration? If so, please revise the sentence.
è We revised the sentence as following “to obtain a difference of Hb 1.5 g/dl with an estimated standard deviation of 2.5”. Thank your kindly comments.
The term ‘perioperative’ is frequently used for either preoperative, intraoperative and/or postoperative period. It would seem that the iron in this study was actually given intraoperatively. If that is the case, perhaps the wording should be changed accordingly. We changed the term ‘perioperative’ to ‘intraoperative’ in this manuscript. Thus, our title became an “Effect of intraoperative ferric carboxymaltose in joint arthroplasty patients” a randomized trial”.
2nd Reviewer
The authors have presented a trial of intra operative FCM administration in large joint arthroplasty (TKR, THR) cases. Although a small study, the results are consistant with previous publications.
The limitations of the study presented by the authors should include the 'lack thereof' iron deficiency or iron deficiency anemia (ID, IDA) in the cohorts that were included in the trial. Blinding is another very difficult design but should also be addressed. This study did not evaluate whether patients has already iron deficiency or not when screening and enrollment stage. We excluded patients with Hb < 10 g/dl, which seemed to exclude most patients with iron deficiency. In Table 3, our subjects showed identical rate of absolute iron deficiency in preoperative stage, thus, it may not influence of our results. Nevertheless, it may be one of limitation of current study.
Another limitation is the explanation of the high ferritin levels that could have been helped if CRP levles would have been done.
è Acute phase reaction after operation cause an increase in serum ferritin and C-reactive protein (CRP). To diagnosis of functional iron deficiency anemia when CRP > 5 mg/dl, ferritin level must be over 100 ng/ml. We mentioned this definition in footnote of Table 3 and further added preoperative /postoperative day 5 CRP level in Table 3.
Last, this study, if conducted inpatient with ID or IDA would have resulted in statistically significant outcomes. We agree with the reviewer’s opinion that the results may statistically more significant in patients with preoperative iron deficiency. According to Khalafallah et al,they screened patients with iron deficiency anemia at postoperative day1, then gave a single FCM. It caused improve of mean Hb level after 4 weeks. Our Group C showed significantly higher portion of functional iron deficiency state after surgery than patient of Group F. The current study demonstrated that intraoperative IV iron treatment may prevent or reduce to become an iron deficiency state even in patients without preoperative iron deficiency. Furthermore, like Khalafallah’ study, it showed the improvement of Hb recovery after 1 month. If we included all patients with preoperative iron deficiency anemia, it would have been yield different results. We added these sentences and modified one paragraph within line 251-256. There is question that patient who lose operative blood are losing iron. Early supplementation (after induction of anesthesia, regional or genral) might help in early recovery. Early Recovery After Surgery (ERAS) is becoming recognized as an effective way of reducing LOS, complication and improving both fuctional and emotial quality of life. As such, the author's address this study's limits of addressing this phenomenon but nonetheless, speculation as to how early iron replenishment might help would improve the discussion.
è Aspect of ERAS, IV iron treatment might help in early recovery. Evaluation how early (intra- or immediate postoperative) iron replenishment might help ERAS protocol related with quality of life in orthopedic surgery. We added you remarks in the limitation section, and now it reads, A study with larger population will be needed to further evaluate how short-term administration of ferric carboxymaltose might help in early recovery, length of hospital stay, clinically related outcomes including quality of life, serious adverse events and safety parameters.
A few structural comments:
P1, line 34 - Suggest replacing "required" with "results in"
P1 line 37 - Suggest replacing "side effects" with "adverse effects".
Thank you for your kind remarks for our manuscript. We revised these words as your comments.
P2 line 46, 47 - would remove this sentence.
We removed following sentence; “These uncertainties about exact operation day may further hinder the PBM in outpatient clinics”.

Reviewer 2 Report
The authors have presented a trial of intra operative FCM administration in large joint arthroplasty (TKR, THR) cases. Although a small study, the results are consistant with previous publications.
The limitations of the study presented by the authors should inlcude the 'lack thereof' iron deficiency or iron deficiency anemia (ID, IDA) in the cohorts that were included in the trial. Blinding is another very diffecult design but should also be addressed. Another limitation is the explanation of the high ferritin levels that could have been helped if CRP levles would have been done. Last, this study, if conducted inpatient withID or IDA would have resulted in statistically significant outcomes.
There is question that patient who lose operative blood are losing iron. Early supplimentation (after induction of anesthesia, regional or genral) might help in early recovery. Early Recovery After Surgery (ERAS) is becoming recognized as an effective way of reducing LOS, complication and improving both fuctional and emotial quality of life. As such, the author's address this study's limits of addressing this phenomanon but non the less, speculation as to how early iron replenishment might help would improve the discussion.
A few structural comments:
P1, line 34 - Suggest replacing "required" with "results in"
P1 line 37 - Suggest replacing "side effects" with "adverse effects".
P2 line 46, 47 - would remove this sentence.
Author Response
The authors have presented a trial of intra operative FCM administration in large joint arthroplasty (TKR, THR) cases. Although a small study, the results are consistant with previous publications.
The limitations of the study presented by the authors should include the 'lack thereof' iron deficiency or iron deficiency anemia (ID, IDA) in the cohorts that were included in the trial. Blinding is another very difficult design but should also be addressed. This study did not evaluate whether patients has already iron deficiency or not when screening and enrollment stage. We excluded patients with Hb < 10 g/dl, which seemed to exclude most patients with iron deficiency. In Table 3, our subjects showed identical rate of absolute iron deficiency in preoperative stage, thus, it may not influence of our results. Nevertheless, it may be one of limitation of current study.
Another limitation is the explanation of the high ferritin levels that could have been helped if CRP levles would have been done.
è Acute phase reaction after operation cause an increase in serum ferritin and C-reactive protein (CRP). To diagnosis of functional iron deficiency anemia when CRP > 5 mg/dl, ferritin level must be over 100 ng/ml. We mentioned this definition in footnote of Table 3 and further added preoperative /postoperative day 5 CRP level in Table 3.
Last, this study, if conducted inpatient with ID or IDA would have resulted in statistically significant outcomes. We agree with the reviewer’s opinion that the results may statistically more significant in patients with preoperative iron deficiency. According to Khalafallah et al,they screened patients with iron deficiency anemia at postoperative day1, then gave a single FCM. It caused improve of mean Hb level after 4 weeks. Our Group C showed significantly higher portion of functional iron deficiency state after surgery than patient of Group F. The current study demonstrated that intraoperative IV iron treatment may prevent or reduce to become an iron deficiency state even in patients without preoperative iron deficiency. Furthermore, like Khalafallah’ study, it showed the improvement of Hb recovery after 1 month. If we included all patients with preoperative iron deficiency anemia, it would have been yield different results. We added these sentences and modified one paragraph within line 251-256. There is question that patient who lose operative blood are losing iron. Early supplementation (after induction of anesthesia, regional or genral) might help in early recovery. Early Recovery After Surgery (ERAS) is becoming recognized as an effective way of reducing LOS, complication and improving both fuctional and emotial quality of life. As such, the author's address this study's limits of addressing this phenomenon but nonetheless, speculation as to how early iron replenishment might help would improve the discussion.
è Aspect of ERAS, IV iron treatment might help in early recovery. Evaluation how early (intra- or immediate postoperative) iron replenishment might help ERAS protocol related with quality of life in orthopedic surgery. We added you remarks in the limitation section, and now it reads, A study with larger population will be needed to further evaluate how short-term administration of ferric carboxymaltose might help in early recovery, length of hospital stay, clinically related outcomes including quality of life, serious adverse events and safety parameters.
A few structural comments:
P1, line 34 - Suggest replacing "required" with "results in"
P1 line 37 - Suggest replacing "side effects" with "adverse effects".
Thank you for your kind remarks for our manuscript. We revised these words as your comments.
P2 line 46, 47 - would remove this sentence.
We removed following sentence; “These uncertainties about exact operation day may further hinder the PBM in outpatient clinics”.
